# Projecting COVID-19 isolation bed requirements for people experiencing homelessness

Tanvi A. Ingle[1], Maike Morrison[2], Xutong Wang[1], Timothy Mercer[3,4,5], Vella Karman[6], Spencer Fox[1], Lauren Ancel Meyers[1,7]*

1 Department of Integrative Biology, The University of Texas at Austin, Austin, Texas, United States of America, 2 Department of Biology, Stanford University, Stanford, California, United States of America, 3 Department of Population Health, The University of Texas at Austin Dell Medical School, Austin, Texas, United Staites of America, 4 Department of Internal Medicine, The University of Texas at Austin Dell Medical School, Austin, Texas, United States of America, 5 CommUnityCare Federally Qualified Health Centers, Austin, Texas, United States of America, 6 Homeless Services Division, Austin Public Health, Austin, Texas, United States of America, 7 Santa Fe Institute, Santa Fe, New Mexico, United States of America

* laurenmeyers@austin.utexas.edu

**Data Availability Statement:** All relevant data are within the manuscript and its Supporting Information files.

**Funding:** This study was supported in the form of grants by Centers for Disease Control (CDC U01

## Abstract

As COVID-19 spreads across the United States, people experiencing homelessness (PEH) are among the most vulnerable to the virus. To mitigate transmission, municipal governments are procuring isolation facilities for PEH to utilize following possible exposure to the virus. Here we describe the framework for anticipating isolation bed demand in PEH communities that we developed to support public health planning in Austin, Texas during March 2020. Using a mathematical model of COVID-19 transmission, we projected that, under no social distancing orders, a maximum of 299 (95% Confidence Interval: 223, 321) PEH may require isolation rooms in the same week. Based on these analyses, Austin Public Health finalized a lease agreement for 205 isolation rooms on March 27th 2020. As of October 7th 2020, a maximum of 130 rooms have been used on a single day, and a total of 602 PEH have used the facility. As a general rule of thumb, we expect the peak proportion of the PEH population that will require isolation to be roughly triple the projected peak daily incidence in the city. This framework can guide the provisioning of COVID-19 isolation and post-acute care facilities for high risk communities throughout the United States.

## Introduction

As of December 16th 2020, a new coronavirus (SARS-CoV-2) has emerged into a global pandemic, with more than 16,519,668 confirmed cases of the disease (COVID-19) and 302,992 COVID-19 deaths reported in the United States [1]. In response, 42 states and the District of Columbia imposed stay-at-home orders, with the earliest beginning in mid-March [2]. All have since begun relaxing these orders, with great heterogeneity in duration and the details of continued restrictions across states [3]. During even the strictest stay-home measures, some populations have difficulty avoiding contacts and reducing transmission risks, including those

IP00136) awarded to XW and LAM, and by National Institute of Health (NIH R01 AI151176) awarded to LAM. Tito's Handmade Vodka provided support in the form of salary for SJF. The specific roles of these authors are articulated in the 'author contributions' section. The funders had no role in study design, data collection and analysis, decision to publish, or preparation of the manuscript.

**Competing interests:** The authors have read the journal's policy and have the following competing interests: SJF is financially supported by a generous donation made by Tito's Handmade Vodka to The University of Texas at Austin. There are no patents, products in development or marketed products associated with this research to declare. This does not alter our adherence to PLOS ONE policies on sharing data and materials.

working in healthcare and other essential industries, individuals in correctional and detention facilities, and residents of long-term care facilities [4]. People experiencing homelessness (PEH) are at a particularly high risk. Most lack facilities for self-isolation and quarantine. They often reside in densely packed outdoor encampments or shelters, congregate near donation centers to receive food and other basic needs, and share a few restrooms, handwashing stations, and showers [5]. PEH also suffer from higher rates of comorbidities, accelerated aging and limited access to healthcare services, compounding their risk of COVID-19 hospitalization and mortality [6, 7]. By April 2020, outbreaks of COVID-19 had already begun sweeping through homeless populations in many major cities, such as Boston [8], Seattle [9], and San Francisco [10]. In order to slow clusters of transmission in these communities, the CDC recommends providing isolation facilities for the isolation and quarantine of those exposed to COVID-19 and awaiting testing, those awaiting test results, and those who have tested positive and are recovering [11].

In line with these recommendations, some cities have established COVID-19 isolation facilities in unused convention centers, hotel rooms, and college dormitories [12–15]. A key challenge has been projecting the number of isolation beds that will be required to ensure the health and safety of PEH communities in advance of pandemic waves. The demand will depend on the future SARS-COV-2 transmission dynamics among PEH, the availability, speed and accuracy of testing, and the duration and severity of illness for PEH.

The timing and extent of COVID-19 outbreaks within PEH communities remains uncertain, given the novelty of the virus and undetermined future interventions that could mitigate spread.

To support planning by the Homeless Services Division of Austin Public Health in March 2020, we used a mathematical model of COVID-19 transmission in the Austin-Round Rock metropolitan service area (MSA) to project the number of isolation rooms that will be required to allow PEH to self-isolate while awaiting test results or through the infectious period. In addition, to support decision making in other cities, we share a simple formula to calculate the percentage of PEH who will need an isolation bed as a percentage of peak COVID-19 incidence among the homeless population.

## Methods

Our hierarchical approach to estimating peak demand for PEH isolation beds first uses a compartmental model to project city-wide spread of COVID-19 under a range of intervention scenarios and then uses demographic data on the PEH community to derive estimates for the daily numbers of PEH that will require isolation for each scenario.

### Projecting the first wave of the COVID-19 pandemic in the Austin-Round Rock MSA under various intervention scenarios

We estimated the maximum daily number of COVID-19 isolation beds that will be required to support the city of Austin's PEH population. We use a stochastic age- and risk-structured susceptible-exposed-asymptomatic-symptomatic-hospitalized-recovered (SEAYHR) compartmental model of COVID-19 transmission to project incidence among PEH. This model was derived from previously published models developed early in the pandemic, and assumes an $R_0$ of 2.2, symptomatic proportion of 82.1%, and that the infectiousness of asymptomatic cases is 47% that of symptomatic cases [16]. Using this model, we projected COVID-19 prevalence across the Austin-Round Rock MSA from March 1, 2020 to July 1, 2020 under seven intervention scenarios: (1) no interventions, (2-4) indefinite school closures plus a four-week stay-home order that decreases non-household contacts by either 50%, 75%, or 90%, and (5-7)

indefinite school closures plus a <u>four-month</u> stay-home order that decreases non-household contacts by either 50%, 75%, or 90%. For each scenario, we ran 100 stochastic simulations. We assumed published estimates for COVID-19 transmission rates and age-group and risk-group severity. The model is described in full detail in S6 Fig, S1–S7 Tables in S2 Appendix.

## Projecting COVID-19 prevalence within PEH population

We translate each population-wide projection into an estimate for the daily number of PEH who become infected with COVID-19, assuming that the prevalence in the PEH community will mirror the corresponding age-specific prevalence across the city as a whole.

For each simulated scenario, we derive a daily time series of PEH requiring an isolation bed while awaiting COVID-19 test results or while recovering from a test-confirmed case of COVID-19. Let $I(i,t)$ denote the projected number of new COVID-19 infections in age group $i$ on day $t$ in a given stochastic simulation. We estimate the corresponding number new infections in the PEH community as

$$I_h(t) = N_h \cdot \sum_{i \in a} \frac{I(i,t)}{N_A} \cdot p_h(i)$$

where $N_A$ is the population size in Austin, $N_h$ the estimated PEH population size in Austin, the summation is taken over the five age groups $a$ = {0-4, 5-17, 18-49, 50-64, 65+}, and $p_h(i)$ is the estimated proportion of the local PEH population in age group $i$ (Table 1). To account for undercounting [17], we set $N_h$ equal to 1.4 times the point-in-time unsheltered PEH population size estimate (provided by the Homeless Services Division of Austin Public Health) plus the sheltered PEH population.

## Estimating isolation bed demand

We estimate the number of beds needed on a given day, $B(t)$, using estimates of the total number of tests conducted on previous days, the proportion of those tests that are positive, and the respective durations of isolation for those testing positive or negative:

$$B(t) = \sum_{\tau=t-t_{neg}}^{t} T(\tau) \cdot (1 - p(\text{infected}|\text{tested})) + \sum_{\tau=t-t_{pos}}^{t} T(\tau) \cdot p(\text{infected}|\text{tested})$$

where $B(t)$ is the number of beds required on day $t$, $T(\tau)$ is the number of tests conducted on day $\tau$, and $t_{neg}$ and $t_{pos}$ are the durations that individuals who test negative and positive respectively must remain in isolation. In our model we assume perfect test sensitivity and specificity, so $p(\text{infected}|\text{tested}) = p(\text{positive}|\text{tested})$ and corresponds to the proportion of tests that return a positive result. The first term in this equation corresponds to individuals who test negative and remain in isolation for $t_{neg}$ days, while the second term correspond to individuals who test positive and remain in isolation for a longer period of $t_{pos}$ days.

**Table 1. Estimated age distribution of PEH in Austin, TX [18].**

| Age Groups | Percent of Austin PEH Population in age group ($p_h(i)$) |
|---|---|
| 0-4 years | 0.3% |
| 5-17 years | 0.7% |
| 18-49 years | 60% |
| 50-64 years | 36% |
| 65+ years | 3% |

Every person who is tested is assumed to immediately enter isolation. We assume an individual will remain in isolation until they receive a negative test result ($t_\text{neg}$ days), or depart as a result of recovery or hospitalization ($t_\text{pos}$ days).

We express the number of tests administered on a given day, $T(t)$, as $T(t) = N_h \cdot p_\text{tested}(t)$, where $p_\text{tested}(t)$ is the probability of a PEH being tested on day $t$ and, as before, $N_h$ is the estimated PEH population size. Using Bayes theorem, we can compute the probability of a PEH being tested on day $t$ as $p_\text{tested}(t) = p_\text{infected}(t) \cdot \frac{p(\text{tested}|\text{infected})}{p(\text{infected}|\text{tested})}$. Substituting this definition of $p_\text{tested}(t)$ into the equation for $T(t)$ gives that $T(t) = N_h \cdot p_\text{tested}(t) = N_h \cdot p_\text{infected}(t) \cdot \frac{p(\text{tested}|\text{infected})}{p(\text{infected}|\text{tested})}$. Further, we note that $N_h \cdot p_\text{infected}(t) = I_h(t-k)$, where $k$ is the time delay between infection onset and testing, and so $I_h(t-k)$ is the number of PEH who became infected $k$ days ago. Making this substitution gives our final equation for the number of tests on a given day,

$$T(t) = I_h(t - k) \cdot \frac{p(\text{tested}|\text{infected})}{p(\text{infected}|\text{tested})}$$

where $p(\text{tested}|\text{infected})$ is the probability an infected individual is tested and $p(\text{infected}|\text{tested})$ is the probability a tested individual is infected. Here, $p(\text{tested}|\text{infected})$ and $p(\text{infected}|\text{tested})$ are parameters we estimate from the literature (Table 2), while $I_h(t-k)$ is computed from our stochastic epidemic projections, as discussed above.

We can combine these equations to provide a conservative (i.e., slightly overestimated) projection for the required number of isolation beds. The following *rule of thumb* assumes that the pandemic wave hovers around its maximum daily incidence for two weeks:

$$B_\text{max} = t_\text{neg} \cdot (1 - p(\text{infected}|\text{tested})) + t_\text{pos} \cdot p(\text{infected}|\text{tested})) \cdot \max(T(t))$$

where $B_\text{max}$ is the maximum proportion of the PEH that will require isolation beds on any day in the pandemic and $\max(T(t)) = p(\text{tested}|\text{infected})/p(\text{infected}|\text{tested}) \cdot \max(I(h))$. Thus, for the parameters given in Table 2, $B_\text{max} = 3.24 \cdot \max(I_h(t))$.

For each intervention scenario implemented in our epidemic projections, we identify the average and 95% confidence interval for maximum bed requirements across the 100 stochastic realizations and the 7 scenarios. We used parameter estimates for model inputs from local data wherever possible (Table 2), and conducted sensitivity analyses for chosen parameters to better understand their impact on the model (S1–S5 Figs in S1 Appendix).

## Results

Isolation bed requirements are expected to depend on the efficacy of intervention orders (Fig 1, Table 3). Intuitively, measures that slow spread and depress the pandemic peak will reduce

**Table 2. Model parameters.**

| Parameter | Parameter description and justification | Value | Source |
|---|---|---|---|
| $p(\text{tested}|\text{infected})$ | Probability that an infected individual is tested. Based on the estimates for COVID-19 case detection probability. | 0.1 | [19] |
| $p(\text{infected}|\text{tested})$ | Probability that a tested individual is infected, is based on Texas' test positive rates at the time of analysis. | 0.098 | [20] |
| $k$ | Duration an infected individual waits before getting tested | 5 days | [21] |
| $t_{pos}$ | Duration a positive individual spends in isolation based on CDC guidelines. | 14 days | [4] |
| $t_{neg}$ | Duration a negative individual spends in isolation before receiving results and being cleared of the virus. | 2 days | [22] |

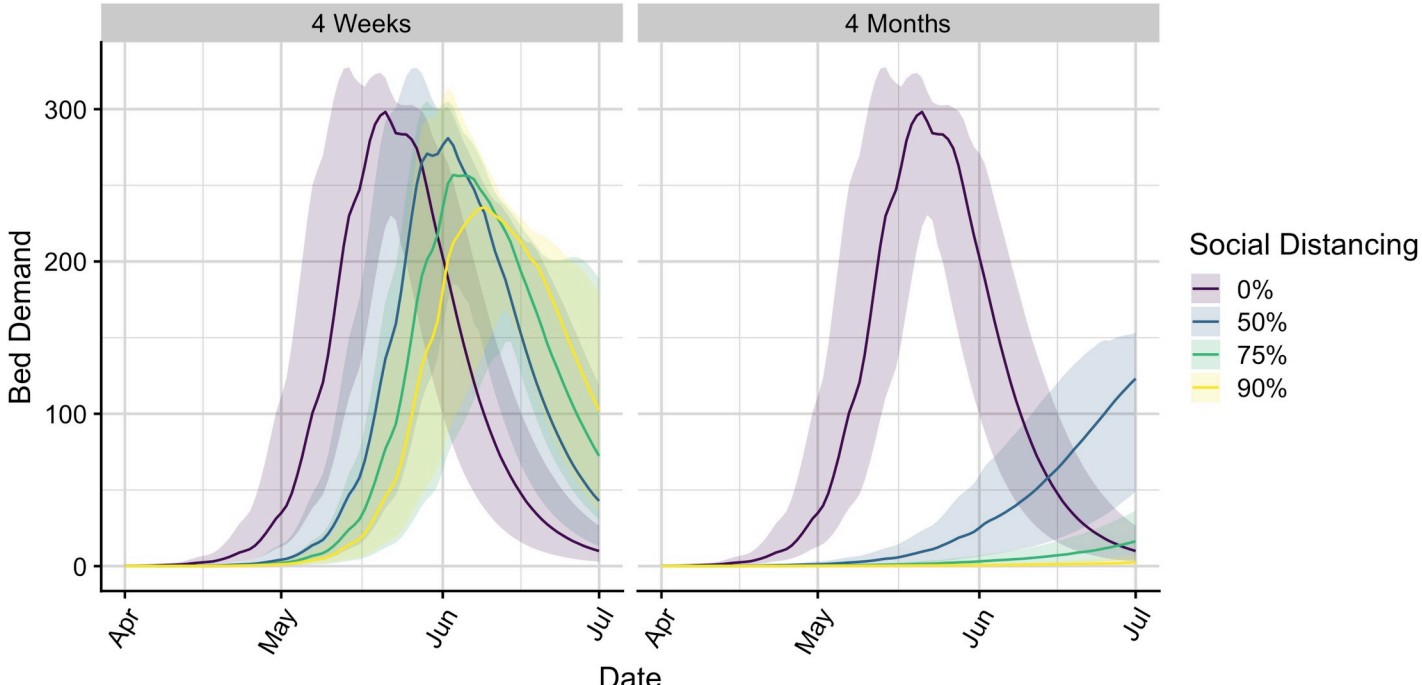

**Fig 1. Projections of daily PEH isolation bed demand with 95% prediction intervals under seven intervention scenarios.** Stay-home orders are assumed to hold for either four weeks (left) or four months (right). Color indicates the reduction in non-household transmission during the intervention period, with 0% indicating no mitigation and other values indicating a combination of school closures and the specified reduction in contacts outside the home. Bed demand is the estimated number of PEH that will require isolation either while waiting for SARS-CoV-2 diagnostic test results or following a positive test result.

the number of PEH isolation beds required through July 1, 2020 (Table 1). Under the worst-case intervention scenario considered–50% effective social distancing for only four weeks–the model projects that more than 281 (176-302) isolation beds will be required by early June. The best-case scenario projects that only a few beds would be required throughout the period. These projections are highly sensitive to our parameter assumptions (S1–S5 Figs in S1 Appendix), particularly the testing effort ($p$(tested|infected) and the time between testing and receiving results ($t_{neg}$). The city may require over 1000 isolation beds if the proportion of infected individuals tested exceeds 0.3, or if the average lag between testing and receiving results exceeds nine days (S1 and S5 Figs in S1 Appendix).

To allow extrapolation from Austin to other US cities, we derive a conservative rule of thumb that relates the peak isolation bed needs ($B_{max}$) to the projected peak daily incidence ($I_{max}$) in the city, as given by $B_{max} = 3.24 \cdot I_{max}$. As validation, we regressed the maximum

**Table 3. Maximum isolation bed demand and date of peak with 95% confidence intervals over seven intervention scenarios.**

| Duration of Intervention | Contact Reduction | Maximum Bed Demand | Date of Peak Bed Demand |
|---|---|---|---|
| No Intervention | 0% | 299 (223-321) | May 21 (May 14 - May 28) |
| 4 Weeks | 50% | 281 (176-302) | June 2 (May 26 - June 13) |
| | 75% | 257 (82-302) | June 3 (May 29 - June 24) |
| | 90% | 236 (142-266) | June 9 (June 2 - June 22) |
| 4 Months | 50% | 124 (50-154) | July 1 (June 29 - July 1) |
| | 75% | 17 (5-37) | July 1 (July 1 - July 1) |
| | 90% | 3 (1-7) | July 1 (June 30 - July 1) |

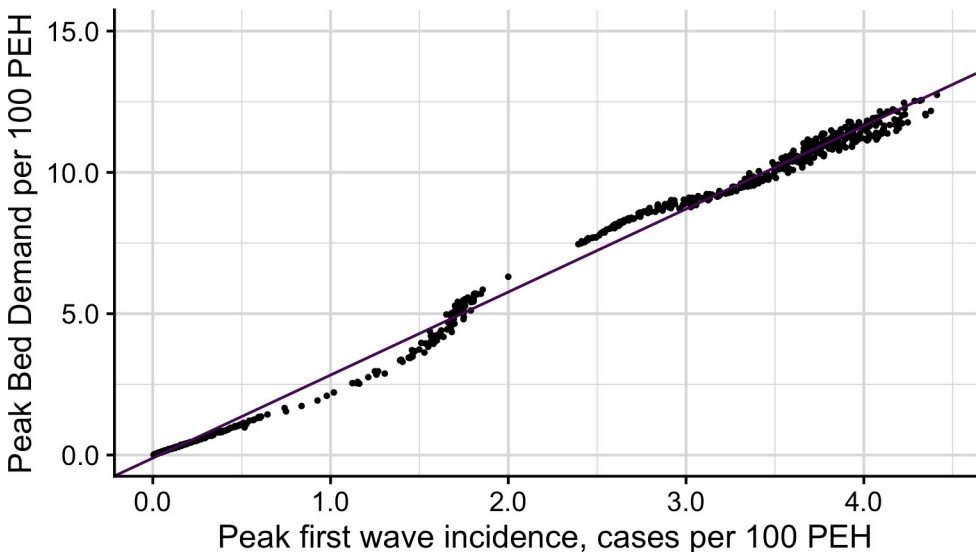

**Fig 2. Relationship between peak COVID-19 case incidence and peak isolation bed requirements per 100 PEH.**
We estimated these values, for each of the 700 total stochastic simulations (100 per each of the 7 scenarios). We fit a linear model and found $y = 2.938 \cdot x - 0.001$, where $x$ is the peak first wave COVID-19 incidence, cases per 100 PEH and $y$ is the peak bed requirement per 100 PEH (See S8 Table in S2 Appendix for regression table). This model can be used to estimate the PEH isolation bed requirements for any city, assuming the city's total PEH population and the estimated peak prevalence of COVID-19 in PEH are known.

number of PEH requiring isolation in each simulation on the maximum daily incidence among PEH, across all seven intervention scenarios, and found that the peak bed requirements were roughly triple the peak incidence (Fig 2). The relationship between peak incidence and PEH isolation needs is sensitive to all parameters in Table 2 except for the delay between infection onset and testing (S6 Fig, S1–S7 Tables in S2 Appendix). Given the COVID-19 surge and increased proactive testing and contact tracing regimes in the summer, we compare our baseline scenario (10% positivity rate and 9.8% detection rate) to one with a testing positivity rate of 19.7% and assume that 20% of infections were detected as was estimated in July [23, 24]. Under these conditions, we project increased peak bed needs for the PEH population, rising from roughly triple to roughly quadruple the peak incidence (S7 Fig in S2 Appendix).

## Discussion

The COVID-19 pandemic in the United States has triggered efforts to protect populations of people experiencing homelessness (PEH). In Austin, Texas, the Parks and Recreation Department partnered with Austin Public Health to open personal hygiene stations around the city, the Office of Sustainability established a micro-food distribution system to prevent long lines at soup kitchens, homeless shelters have instituted CDC-recommended infection screening and prevention protocols, and community outreach partners are leading initiatives to educate PEH about COVID-19 prevention strategies and implement ongoing symptom-based and surveillance testing among this population [25, 26].

In March 2020, the Homeless Services Division of the Austin Public Health Department sought to project the number of rooms that would be needed to isolate potentially exposed members of the PEH community while they waited for test results and, if positive, throughout their infectious period. Our analyses projected, intuitively, that city-wide mitigation efforts that effectively reduced the rate of transmission and peak incidence would likewise reduce the

number of isolation beds that would be required. To be conservative, we derived recommendations under a worse-case scenario in which the March 2020 stay-home order reduced non-household transmission by only 50% for only four weeks. We anticipated that the city will need at least 250 isolation beds through July 1st, 2020. However the demand could exceed 1000 under alternative plausible parameter combinations. The expected demand for isolation beds increases not only with the projected epidemic intensity, but also with the aggressiveness of testing, contact tracing, and isolation efforts in the PEH community. Based on these analyses, on March 27th, 2020, the City of Austin leased an isolation facility with 205 rooms for individuals without a location to safely isolate or quarantine, including essential workers and PEH tested for COVID-19 [26]. Since then, occupancy in this facility has fluctuated dramatically, from 10 to 130 occupants, with PEH making up the majority of occupants. After comparing our projections with the reported COVID-19 hospitalizations in Austin through July 1st, we find that the data most closely match the four month intervention scenarios of 75% and 90% contact reduction, although our model did not incorporate the large fluctuations in transmission actually experienced throughout this period (S10 Table in S2 Appendix) [27].

We provide a ballpark relationship between peak incidence and peak PEH isolation bed needs. Assuming the parameters estimated for Austin's PEH community, the maximum proportion requiring isolation on any day of the pandemic is roughly three times the peak daily incidence of the pandemic in the city as a whole. When we scale our results from Austin to Los Angeles, California, we estimate that the city would need just over 6,400 isolation beds during a completely uncontrolled pandemic. Los Angeles public health officials reserved 4,117 isolation beds for the population which is roughly two thirds of our peak estimate. Interestingly, the City of Austin also chose to reserve roughly two thirds of our peak estimate of 299 isolation beds in an uncontrolled pandemic [28].

We highlight three key assumptions that may not hold in Austin or elsewhere. First, we assume that COVID-19 spreads at the same rate within the PEH community as it does in the general population and that social distancing interventions are equally effective at reducing transmission within this community. However, considering that PEH generally reside in densely packed shelters or encampments with limited access to sanitation that make it difficult to self-isolate, congregate in facilities that deliver essential services, have higher rates of underlying medical conditions and are more likely to fall ill, it is highly probable that PEH have higher levels of transmission and susceptibility to COVID-19 [4, 29, 30] than the general population. While public health officials in Austin have enacted various efforts to protect this vulnerable population, like acquiring five protective lodging facilities to help high risk individuals socially distance, it is currently unknown whether these measures have successfully counterbalanced these risks. In the absence of effective social distancing measures in the PEH community, we would expect that disease transmission would be amplified and possibly cause explosive clusters of cases. Pandemic waves among PEH might thus resemble our projections that assume no interventions, even when the rest of the city is isolating, resulting in earlier and higher peaks in isolation needs.

Second, our estimates are highly sensitive to the speed of COVID-19 testing for PEH. We assume that roughly 10% of PEH infections would be tested ($p$(tested|infected)), given CDC national estimates for case detection rates [4]. We chose to hold this parameter constant, given the limited information about future testing rates among PEH at the time the projections were made, but find that isolation bed needs likely increased during the July surge experienced in Texas (S7 Fig in S2 Appendix). Under a proactive contact tracing scenario in which a greater proportion of cases would be tested, far more isolation beds would be required (S1 Fig in S1 Appendix). In fact the COVID-19 screening program run by Dell Medical School, CommUnity Care clinic, and Austin Public Health, tested 634 PEH for COVID-19 by November 14th

2020. All the PEH who tested positive and were unable to safely self-isolate were referred to the City's isolation facility.

Finally, our framework assumes perfect test sensitivity and specificity, based on the high reported accuracy of the PCR tests that were available at the time of the analysis [31]. However, there are now many testing options including the possibility for frequent, low cost rapid antigen tests that are considerably less sensitive [32]. Our framework can be adapted to project isolation bed needs under these different testing regimes, and our sensitivity analysis with respect to testing rates illustrates their non-linear impact on isolation bed needs (S1, S2 Figs in S1 Appendix).

This simple framework is designed to inform real-time provisioning of costly resources to protect vulnerable populations in the face of an epidemic threat with a high degree of uncertainty [33]. By coupling COVID-19 projections with PEH demographic information, cities like Austin can make cost-effective decisions as new pandemic waves rise in the months ahead [34, 35].

## Supporting information

**S1 Appendix. Sensitivity analyses for isolation bed demand estimates.**
(DOCX)

**S2 Appendix. COVID-19 transmission model.**
(DOCX)

## Acknowledgments

The authors acknowledge the Texas Advanced Computing Center (TACC) at The University of Texas at Austin for providing HPC, visualization, database, and grid resources that have contributed to the research results reported within this paper. URL: http://www.tacc.utexas.edu

## Author Contributions

**Conceptualization:** Tanvi A. Ingle, Maike Morrison, Spencer Fox.

**Data curation:** Tanvi A. Ingle, Maike Morrison, Xutong Wang, Timothy Mercer, Vella Karman.

**Formal analysis:** Tanvi A. Ingle, Maike Morrison, Xutong Wang.

**Funding acquisition:** Lauren Ancel Meyers.

**Investigation:** Tanvi A. Ingle, Maike Morrison, Xutong Wang, Lauren Ancel Meyers.

**Methodology:** Tanvi A. Ingle, Maike Morrison.

**Project administration:** Spencer Fox, Lauren Ancel Meyers.

**Resources:** Lauren Ancel Meyers.

**Supervision:** Spencer Fox, Lauren Ancel Meyers.

**Validation:** Spencer Fox, Lauren Ancel Meyers.

**Visualization:** Tanvi A. Ingle.

**Writing – original draft:** Tanvi A. Ingle, Maike Morrison, Xutong Wang, Spencer Fox.

**Writing – review & editing:** Tanvi A. Ingle, Xutong Wang, Timothy Mercer, Vella Karman, Spencer Fox, Lauren Ancel Meyers.

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
