## [Decision Letter · Decision Letter 0]

20 Jan 2021

PONE-D-20-39557

Projecting COVID-19 Isolation Bed Requirements for People Experiencing Homelessness

PLOS ONE

Dear Dr. Meyers,

Thank you for submitting your manuscript to PLOS ONE. After careful consideration, we feel that it has merit but does not fully meet PLOS ONE’s publication criteria as it currently stands. Therefore, we invite you to submit a revised version of the manuscript that addresses the points raised during the review process.

We look forward to receiving your revised manuscript.

Kind regards,

Martial L Ndeffo Mbah, Ph.D

Academic Editor

PLOS ONE

Journal Requirements:

2.) Thank you for stating the following financial disclosure:

'The funders had no role in study design, data collection and analysis, decision to publish, or preparation of the manuscript.'

3.) We note you have included a table to which you do not refer in the text of your manuscript. Please ensure that you refer to Table 3 in your text; if accepted, production will need this reference to link the reader to the Table.

Reviewers' comments:

Reviewer's Responses to Questions

**Comments to the Author**

1. Is the manuscript technically sound, and do the data support the conclusions?

Reviewer #1: Yes

Reviewer #2: Yes

2. Has the statistical analysis been performed appropriately and rigorously? 

Reviewer #1: Yes

Reviewer #2: Yes

3. Have the authors made all data underlying the findings in their manuscript fully available?

Reviewer #1: Yes

Reviewer #2: Yes

4. Is the manuscript presented in an intelligible fashion and written in standard English?

Reviewer #1: Yes

Reviewer #2: Yes

5. Review Comments to the Author

Reviewer #1: The authors developed a stochastic age- and risk- structured SEIR-like compartmental model of COVID-19 transmission in the Austin to project the number of isolation rooms that will be required to allow people experiencing homelessness to self-isolate while awaiting test results or through the infectious period. The authors projected the COVID-19 prevalence in Austin under seven reasonable intervention scenarios and sensitivity analysis for important parameter were done. Moreover, the authors showed that peak demand for PEH isolation bed to be roughly triple the projected peak daily incidence in the city, which is useful and easy for supporting decision making in other cities.

The manuscript overall is well presented. The idea behind the work holds relevance and is of importance to public health. Some issues can be addressed to increase the clarity of this manuscript.

1. The main result (peak demand for PEH isolation bed) is highly dependent on the projected peak daily incidence in the city. I am wondering whether their model are calibrated and validated by the available data in Austin. If not, I would suggest that the authors at least to demonstrate some results regarding their models projection (e.g., projected daily incidence) in the manuscript or SI.

2. The authors assumed perfect test sensitivity and specificity i.e., p(infected|tested)=p(positive|tested), and I am wondering whether this assumption would greatly affect the results or not. Please clarify.

As the authors stated in the manuscript, their estimates are highly sensitive to the speed of COVID-19 testing and truly infection prevalence for PEH. At early stage (March to July), the detection/test rates are quite low. In the future, under a more strict quarantine/contact tracing policy, far more isolation beds would be required. However, the work did provide a nice framework to inform real-time provisioning to protect vulnerable PEH populations.

Reviewer #2: Dear Authors,

I think you address an important aspect of pandemic management in a population that often overlooked. The writing and presentation of the work is solid and the methods are appropriate.

A few remarks:

1. I am confused by the parameter k. According to figure S3, there is no difference in bed demand in the first 4 weeks past simulation start if someone gets infected and waits 2 days to get tested (and start isolating) or if someone waits 10 days to get tested and start isolating. Over the first 4 months, it appears that waiting longer to get tested (and begin isolating) slightly decreases the total number of beds needed. This seems counterintuitive to me, given that an infected person would presumably infect many more people over 10 days than over 2.

2. In Figure 2, incidence is listed as a percentage, which is quite unusual. Is this the percentage of the PEH population that is infected? Should this be cases per 100 persons? I don't know how to interpret this value.

3. The ongoing surge of COVID-19 in Texas probably changes suitable parameters of this model quite a bit (in recent weeks, the highest percentage of positive tests was 22%, more than double the assumed value here), and might make the overall rule of thumb less applicable. I realize that this is a situation in constant flux, but I think it would be useful to address, even briefly, how projections are different now than they were in July. How many isolation beds are in use in Austin? How many are needed?

My few suggestions to improve the readability of the manuscript:

1. I would consider moving some of the tables S2 and S3 from Appendix 2 material into the Methods section, so I don't have to immediately dig for the supplemental materials to see how you structured the model.

2. On lines 134 and 137, N(sub)H is very hard to read and looks (in the .pdf) like N(sub)pi. A different font for the formula might make this easier to distinguish.

6. PLOS authors have the option to publish the peer review history of their article (what does this mean?). If published, this will include your full peer review and any attached files.

Reviewer #1: **Yes: **Qimin Huang

Reviewer #2: No

---

## [Author Response · Author response to Decision Letter 0]

20 Apr 2021

Journal, Comment 1

Author Response:

We have revised the manuscript to fit PLOS ONE’s style requirements and have followed the appropriate file naming conventions. Specifically, we have moved all tables directly beneath their first paragraph of mention and corrected the file names.

----------------

Journal, Comment 2

Thank you for stating the following financial disclosure 'The funders had no role in study design, data collection and analysis, decision to publish, or preparation of the manuscript.' At this time, please address the following queries:

Please clarify the sources of funding (financial or material support) for your study. List the grants or organizations that supported your study, including funding received from your institution. State what role the funders took in the study. If the funders had no role in your study, please state: “The funders had no role in study design, data collection and analysis, decision to publish, or preparation of the manuscript." If any authors received a salary from any of your funders, please state which authors and which funders. If you did not receive any funding for this study, please state: “The authors received no specific funding for this work.” Please include your amended statements within your cover letter; we will change the online submission form on your behalf.

Author Response:

This study was supported in the form of grants by Centers for Disease Control (CDC U01 IP00136) awarded to XW and LAM, and by National Institute of Health (NIH R01 AI151176) awarded to LAM. Tito's Handmade Vodka provided support in the form of salary for SJF. The specific roles of these authors are articulated in the ‘author contributions’ section. The funders had no role in study design, data collection and analysis, decision to publish, or preparation of the manuscript.

Salary support was provided to SJF by Tito’s Handmade Vodka, and to XW from CDC U01 IP00136-01-01. 

Competing Interest Statement: 

The authors have read the journal’s policy and have the following competing interests: SJF is financially supported by a generous donation made by Tito's Handmade Vodka to The University of Texas at Austin. There are no patents, products in development or marketed products associated with this research to declare. This does not alter our adherence to PLOS ONE policies on sharing data and materials

We have included the amended statements in the cover letter. 

----------------

Journal, Comment 3

We note you have included a table to which you do not refer in the text of your manuscript. Please ensure that you refer to Table 3 in your text; if accepted, production will need this reference to link the reader to the Table.

Author Response:

We have modified our results section to include a reference to Table 3. 

----------------

Reviewer 1, Point 1

The authors developed a stochastic age- and risk- structured SEIR-like compartmental model of COVID-19 transmission in the Austin to project the number of isolation rooms that will be required to allow people experiencing homelessness to self-isolate while awaiting test results or through the infectious period. The authors projected the COVID-19 prevalence in Austin under seven reasonable intervention scenarios and sensitivity analysis for important parameter were done. Moreover, the authors showed that peak demand for PEH isolation bed to be roughly triple the projected peak daily incidence in the city, which is useful and easy for supporting decision making in other cities. The manuscript overall is well presented. The idea behind the work holds relevance and is of importance to public health. Some issues can be addressed to increase the clarity of this manuscript.

Author Response: 

Thank you for the kind summary of our work, and we appreciate the constructive feedback. 

Reviewer 1, Point 2

The main result (peak demand for PEH isolation bed) is highly dependent on the projected peak daily incidence in the city. I am wondering whether their model are calibrated and validated by the available data in Austin. If not, I would suggest that the authors at least to demonstrate some results regarding their models projection (e.g., projected daily incidence) in the manuscript or SI.

Author Response: 

Thank you for this suggestion. Since the City of Austin does not have a comprehensive dataset on COVID-19 incidence among people experiencing homelessness, we instead created a wide-range of plausible scenarios that showed the range of possible isolation bed requirements. This allowed our model to be used early in the pandemic to support initial public health decision-making. Although Austin’s trajectory did not follow any of our seven initial scenarios exactly, due to fluctuations in transmission rate over the course of the pandemic, we observed at our projected hospitalizations at 75% and 90% contact reduction over a four month period was closest to observed hospitalization data. We have now provided this comparison between our projected and observed hospitalization data (Supplemental Table S10) as part of the discussion.

Modified Discussion Section

In March 2020, the Homeless Services Division of the Austin Public Health Department sought to project the number of rooms that would be needed to isolate potentially exposed members of the PEH community while they waited for test results and, if positive, throughout their infectious period. Our analyses projected, intuitively, that city-wide mitigation efforts that effectively reduced the rate of transmission and peak incidence would likewise reduce the number of isolation beds that would be required. To be conservative, we derived recommendations under a worse-case scenario in which the March 2020 stay-home order reduced non-household transmission by only 50% for only four weeks. We anticipated that the city will need at least 250 isolation beds through July 1st, 2020. However the demand could exceed 1000 under alternative plausible parameter combinations. The expected demand for isolation beds increases not only with the projected epidemic intensity, but also with the aggressiveness of testing, contact tracing, and isolation efforts in the PEH community. Based on these analyses, on March 27th, 2020, the City of Austin leased an isolation facility with 205 rooms for individuals without a location to safely isolate or quarantine, including essential workers and PEH tested for COVID-19 (1). Since then, occupancy in this facility has fluctuated dramatically, from 10 to 130 occupants, with PEH making up the majority of occupants. After comparing our projections with the reported COVID-19 hospitalizations in Austin through July 1st, we find that the data most closely match the four month intervention scenarios of 75% and 90% contact reduction, although our model did not incorporate the large fluctuations in transmission actually experienced throughout this period (S10 Table) (2).

Reviewer 1, Point 3

The authors assumed perfect test sensitivity and specificity i.e., p(infected|tested)=p(positive|tested), and I am wondering whether this assumption would greatly affect the results or not. Please clarify.

Author Response: 

Thank you for this suggestion. Unfortunately, relaxing this assumption would require information related to testing effort and frequency, and the change in viral load during disease progression. Since these data were not available to us at the time of making projections, we believe this analysis to be beyond the scope of the current paper. We have included this limitation of our study and added the following paragraph to our discussion section in effort to address how imperfect test sensitivity could impact isolation bed demand.

Modified Discussion, new paragraph

Finally, our framework assumes perfect test sensitivity and specificity, based on the high reported accuracy of the PCR tests that were available at the time of the analysis (3). However, there are now many testing options including the possibility for frequent, low cost rapid antigen tests that are considerably less sensitive (4). Our framework can be adapted to project isolation bed needs under these different testing regimes, and our sensitivity analysis with respect to testing rates illustrates their non-linear impact on isolation bed needs (S1-S2 Figs).

Reviewer 1, Point 4

As the authors stated in the manuscript, their estimates are highly sensitive to the speed of COVID-19 testing and truly infection prevalence for PEH. At early stage (March to July), the detection/test rates are quite low. In the future, under a more strict quarantine/contact tracing policy, far more isolation beds would be required. However, the work did provide a nice framework to inform real-time provisioning to protect vulnerable PEH populations.

Author Response:

Thank you for this feedback. We have added a statement in our discussion section to clarify our parameters were fixed due to the limited available information about testing rates at the time of our analysis. We also describe how these parameters may fluctuate over time, and the subsequent impact on projected bed demand.

Modified Discussion section 

Second, our estimates are highly sensitive to the speed of COVID-19 testing for PEH. We assume that roughly 10% of PEH infections would be tested (p(tested|infected)), given CDC national estimates for case detection rates (5). We chose to hold this parameter constant, given the limited information about future testing rates among PEH at the time the projections were made, but find that isolation bed needs likely increased during the July surge experienced in Texas (S7 Fig). Under a proactive contact tracing scenario in which a greater proportion of cases would be tested, far more isolation beds would be required (S1 Fig). In fact the COVID-19 screening program run by Dell Medical School, CommUnity Care clinic, and Austin Public Health, tested 634 PEH for COVID-19 by November 14th 2020. All the PEH who tested positive and were unable to safely self-isolate were referred to the City’s isolation facility.

Reviewer 2, Point 1

I think you address an important aspect of pandemic management in a population that often overlooked. The writing and presentation of the work is solid and the methods are appropriate.

Author Response:

We thank the reviewer for their kind appraisal of our work. We certainly hope our work will support further discussion into pandemic management among vulnerable populations. 

Reviewer 2, Point 2

I am confused by the parameter k. According to figure S3, there is no difference in bed demand in the first 4 weeks past simulation start if someone gets infected and waits 2 days to get tested (and start isolating) or if someone waits 10 days to get tested and start isolating. Over the first 4 months, it appears that waiting longer to get tested (and begin isolating) slightly decreases the total number of beds needed. This seems counterintuitive to me, given that an infected person would presumably infect many more people over 10 days than over 2.

Author Response:

We thank the reviewer for noting this inconsistency. The decrease in the total number of beds for some of the scenarios is an artifact of our decision to run our simulations only until July 1st, capturing the first wave of the pandemic. As the k parameter increases, the curve shifts later into the summer and eventually past July 1st, resulting in a smaller perceived peak bed demand. We note that our quick rule-of-thumb to calculate the maximum bed demand does not depend on this k parameter. We have incorporated this clarification into our S3 Figure description, included below. 

Modified S3 Figure Caption

S3 Fig. Sensitivity analysis for the delay between initial infection and testing (k). The red dashed line indicates the baseline value of 5 days originally assumed. As k the expected peak demand stays flat under the four-week intervention scenario and decreases slightly under the four-month scenario. This decrease is an artifact of our decision to run simulations only until July 1st, capturing the first wave of the pandemic. As k increases, the curve shifts later into the summer, eventually past July 1st resulting in a smaller perceived peak bed demand. We note that our quick rule-of-thumb to calculate the maximum bed demand does not depend on this parameter. 

Reviewer 2, Point 3

In Figure 2, incidence is listed as a percentage, which is quite unusual. Is this the percentage of the PEH population that is infected? Should this be cases per 100 persons? I don't know how to interpret this value.

Author Response:

Thank you for asking for this clarification. We have modified our Figure 2 axes and caption to be in terms of peak COVID-19 cases or peak bed demand per 100 PEH. We include our revised Figure 2 in the uploaded Response to Revisions document and included the modified Figure 2 caption below for your convenience. 

Modified Figure 2 Caption: 

Fig 2. Relationship between peak COVID-19 case incidence and peak isolation bed requirements per 100 PEH. We estimated these values for each of the 700 total stochastic simulations (100 per each of the 7 scenarios). We fit a linear model and found y = 2.938x - 0.001 where x is the peak first wave COVID-19 incidence, cases per 100 PEH and y is the peak bed requirement per 100 PEH (See Table S8 for regression table). This model can be used to estimate the PEH isolation bed requirements for any city, assuming the city’s total PEH population and the estimated peak prevalence of COVID-19 in PEH are known.

Reviewer 2, Point 4

The ongoing surge of COVID-19 in Texas probably changes suitable parameters of this model quite a bit (in recent weeks, the highest percentage of positive tests was 22%, more than double the assumed value here), and might make the overall rule of thumb less applicable. I realize that this is a situation in constant flux, but I think it would be useful to address, even briefly, how projections are different now than they were in July. How many isolation beds are in use in Austin? How many are needed?

Author Response: 

Thank you for raising this valid concern. Since the intention of this model was to provide decisions support early in the pandemic and for the first wave, we held the percentage of positive tests parameter constant throughout our analysis. We recognize this assumption has limitations since the proportion of positive tests changes over the course of the pandemic. To address this we have added supplemental Figure S7 which compares the maximum beds required under our assumed percent positive rate of 9.8% and under the observed positive rate of 19.67% reported by the City of Austin in July (6). In the latter case we also altered p(tested|infected) to reflect the new estimate that 20% of all infected cases are being caught (7). This figure helps compare the bed demands at the beginning and during the peak of the first wave. We have also modified our results section to address how projections differ between March 2020 and July 2020. We state in Introduction, by October 7th 2020 after the first wave, a maximum of 130 rooms had been used on a single day corresponding to approximately 4.8 beds per 100 PEH. 

Modified Results section: 

To allow extrapolation from Austin to other US cities, we derive a conservative rule of thumb that relates the peak isolation bed needs (B_max) to the projected peak daily incidence (I_max) in the city, as given by B_max = 3.24*I_max . As validation, we regressed the maximum number of PEH requiring isolation in each simulation on the maximum daily incidence among PEH, across all seven intervention scenarios, and found that the peak bed requirements were roughly triple the peak incidence (Fig 2). The relationship between peak incidence and PEH isolation needs is sensitive to all parameters in Table 2 except for the delay between infection onset and testing (Fig S6, S1-S7 Tables). Given the COVID-19 surge and increased proactive testing and contact tracing regimes in the summer, we compare our baseline scenario (10% positivity rate and 9.8% detection rate) to one with a testing positivity rate of 19.7% and assume that 20% of infections were detected as was estimated in July (7,8). Under these conditions, we project increased peak bed needs for the PEH population, rising from roughly triple to roughly quadruple the peak incidence (S7 Fig). 

Reviewer 2, Point 5

I would consider moving some of the tables S2 and S3 from Appendix 2 material into the Methods section, so I don't have to immediately dig for the supplemental materials to see how you structured the model.

Author Response:

Thank you for this suggestion. We agree that the methods section should include further details about key parameters used in our SEAYHR model to project COVID-19 incidence among PEH. We’ve chosen not to move tables S2 and S3 from Appendix 2 into the methods section, as we think it would distract from the isolation bed model, which is the main focus of the paper. However, we are open to it if the editor agrees with the reviewer. Instead, we have opted to include a few more sentences in our methods section clarifying the parameters used in the SEAYHR model. We have included the revised methods section below for your convenience. 

Modified Methods Section

We estimated the maximum daily number of COVID-19 isolation beds that will be required to support the city of Austin’s PEH population. We use a stochastic age- and risk-structured susceptible-exposed-asymptomatic-symptomatic-hospitalized-recovered (SEAYHR) compartmental model of COVID-19 transmission to project incidence among PEH. This model was derived from previously published models developed early in the pandemic, and assumes an R0 of 2.2, symptomatic proportion of 82.1%, and that the infectiousness of asymptomatic cases is 47% that of symptomatic cases (9). Using this model, we projected COVID-19 prevalence across the Austin-Round Rock MSA from March 1, 2020 to July 1, 2020 under seven intervention scenarios: (1) no interventions, (2-4) indefinite school closures plus a four-week stay-home order that decreases non-household contacts by either 50%, 75%, or 90%, and (5-7) indefinite school closures plus a four-month stay-home order that decreases non-household contacts by either 50%, 75%, or 90%. For each scenario, we ran 100 stochastic simulations. We assumed published estimates for COVID-19 transmission rates and age-group and risk-group severity. The model is described in full detail in Appendix 2 of our supplemental files (Fig S6, S1-S7 Tables).

Reviewer 2, Point 6:

On lines 134 and 137, N(sub)H is very hard to read and looks (in the .pdf) like N(sub)pi. A different font for the formula might make this easier to distinguish.

Author Response:

Thank you for bringing this to our attention. We have decapitalized all subscripts ‘H’ for improved clarity and consistency of our equations. 

References

1. Shorter C GR. Update on Homelessness Services During Response to COVID-19 [Internet]. [cited 2021 Mar 12]. Available from: http://www.austintexas.gov/edims/pio/document.cfm?id=338058

2. ArcGIS Dashboards [Internet]. [cited 2021 Mar 12]. Available from: https://austin.maps.arcgis.com/apps/opsdashboard/index.html#/39e4f8d4acb0433baae6d15a931fa984

3. Vogels CBF, Brito AF, Wyllie AL, Fauver JR, Ott IM, Kalinich CC, et al. Analytical sensitivity and efficiency comparisons of SARS-CoV-2 RT–qPCR primer–probe sets. Nature Microbiology. 2020 Jul 10;5(10):1299–305.

4. Mina MJ, Parker R, Larremore DB. Rethinking Covid-19 Test Sensitivity - A Strategy for Containment. N Engl J Med. 2020 Nov 26;383(22):e120.

5. CDC. Community, Work, and School [Internet]. 2021 [cited 2021 Mar 12]. Available from: https://www.cdc.gov/coronavirus/2019-ncov/community/correction-detention/guidance-correctional-detention.html

6. ArcGIS Dashboards [Internet]. [cited 2021 Mar 5]. Available from: https://austin.maps.arcgis.com/apps/opsdashboard/index.html#/f8b42573e2df477b87be64bb69574b8e

7. Kalish H, Klumpp-Thomas C, Hunsberger S, Baus HA, Fay MP, Siripong N, et al. Mapping a Pandemic: SARS-CoV-2 Seropositivity in the United States. medRxiv [Internet]. 2021 Jan 31; Available from: http://dx.doi.org/10.1101/2021.01.27.21250570

8. ArcGIS Dashboards [Internet]. [cited 2021 Mar 12]. Available from: https://austin.maps.arcgis.com/apps/opsdashboard/index.html#/f8b42573e2df477b87be64bb69574b8e

9. Wang X, Pasco RF, Du Z, Petty M, Fox SJ, Galvani AP, et al. Impact of Social Distancing Measures on Coronavirus Disease Healthcare Demand, Central Texas, USA. Emerg Infect Dis. 2020 Oct;26(10):2361.

---

## [Editor Report · Decision Letter 1]

21 Apr 2021

Projecting COVID-19 Isolation Bed Requirements for People Experiencing Homelessness

PONE-D-20-39557R1

Dear Dr. Meyers,

We’re pleased to inform you that your manuscript has been judged scientifically suitable for publication and will be formally accepted for publication once it meets all outstanding technical requirements.

Kind regards,

Martial L Ndeffo Mbah, Ph.D

Academic Editor

PLOS ONE
---

## [Editor Report · Acceptance letter]

7 May 2021

PONE-D-20-39557R1 

Projecting COVID-19 Isolation Bed Requirements for People Experiencing Homelessness 

Dear Dr. Meyers:

I'm pleased to inform you that your manuscript has been deemed suitable for publication in PLOS ONE. Congratulations! Your manuscript is now with our production department. 

Kind regards, 

on behalf of

Dr. Martial L Ndeffo Mbah 

Academic Editor

PLOS ONE